# New Geometric Models for Shape Quantification of the Dorsal View in Seeds of *Silene* Species

**DOI:** 10.3390/plants11070958

**Published:** 2022-03-31

**Authors:** José Luis Rodríguez-Lorenzo, José Javier Martín-Gómez, Ángel Tocino, Ana Juan, Bohuslav Janoušek, Emilio Cervantes

**Affiliations:** 1Plant Developmental Genetics, Institute of Biophysics v.v.i, Academy of Sciences of the Czech Republic, Královopolská 135, 612 65 Brno, Czech Republic; rodriguez@ibp.cz (J.L.R.-L.); janousek@ibp.cz (B.J.); 2Instituto de Recursos Naturales y Agrobiología del Consejo Superior de Investigaciones Científicas (IRNASA-CSIC), Cordel de Merinas, 40, 37008 Salamanca, Spain; jjavier.martin@irnasa.csic.es; 3Departamento de Matemáticas, Facultad de Ciencias, Universidad de Salamanca, Plaza de la Merced 1-4, 37008 Salamanca, Spain; bacon@usal.es; 4Departamento de Ciencias Ambientales y Recursos Naturales, University of Alicante, San Vicente, 03690 Alicante, Spain; ana.juan@ua.es

**Keywords:** cardioid, convexity, geometry, model, morphology, oval, seed shape, super-ellipse, symmetry

## Abstract

The description of shape in *Silene* seeds is based on adjectives coined by naturalists in the 19th century. The expressions reniform, dorso plana, and dorso canaliculata were applied in reference to lateral or dorsal views of seeds, but the characters described can be submitted now to an analytical description by quantitative methods, allowing shape quantification and the comparison between species or populations. A quantitative morphological analysis is based on the comparison with geometric models that adjust to the shape of seeds. Morphological analysis of the dorsal view of *Silene* seeds based on geometric models is applied here to 26 seed populations belonging to 12 species. According to their dorsal views, the seeds are classified as convex and non-convex. New geometric models are presented for both types, including figures such as super-ellipses and modified ellipses. The values of *J* index (percent of similarity of a seed image with the model) are obtained in representative seed samples from diverse populations and species. The quantitative description of seed shape based on the comparison with geometric models allows the study of variation in shape between species and in populations, as well as the identification of seeds in *Silene* species. The method is of application to other plant species.

## 1. Introduction

Since the 19th century, the shape of *Silene* seeds in their lateral views has been described as reniform, the word being derived from the Latin rein, meaning kidney, thus reniform means kidney-shaped. However, kidneys are not geometric figures, and their shape is not precisely defined. In consequence, the word “reniform” corresponds to descriptive but not analytical language [1] (p. 269). *Silene* seeds also resemble a cardioid curve, and, in contrast to reniform, the expression “cardioid curve” belongs to analytical language because it defines the figure with the precision of an algebraic equation, allowing the unequivocal representation of the figure and the quantification of the degree of similarity in different images resembling it.

We recently proposed a method to describe and quantify seed shape based on the comparison of the seed images with geometric models. The method is based on a measurement called *J* index that gives the percent of similarity between two images: the seed and a model. To obtain *J* index, a geometric figure selected as a model is superimposed on the image of a well-oriented seed searching for a maximum of similarity between both, the seed image and the model [2,3]. Departing from studies in the model plants *Arabidopsis thaliana* (L.) Heynh., *Lotus japonicus* (Regel) K. Larsen, and *Medicago truncatula* Gaertn [4,5,6], morphological descriptions based on geometric models have been applied to seeds in diverse taxa and families, such as *Capparis spinosa* L. (Capparaceae) [7], *Rhus tripartita* (Ucria) Grande (Anacardiaceae) [8], *Jatropha curcas* L. and *Ricinus communis* L. (Euphorbiaceae) [9,10], as well as *Triticum aestivum* L. (Poaceae) [11]. Reviews of seed morphology based on this method have been applied to the families Arecaceae [12], Cactaceae [13], and Vitaceae [14], and general overviews of the subject were presented in the orders Cucurbitales [15], and Ranunculales [16].

The application of geometric models to *Silene* seeds gives new light on questions raised by taxonomists of the 19th century. Boissier applied the term “renifom” often in reference to the lateral view *Silene* seeds (Semina albuminosa, vel reniformia embryone periphaerico …) ([17], p. 351), and similarly did Rohrbach [18]. The seeds of *Silene* in their lateral views resemble a cardioid. Other geometric figures related to the cardioid may adapt better than the cardioid to the shape of seeds in particular species depending, for example, on whether the region close to the hilum is more open or closed, or if the seeds are more rounded or elongated. This allows a quantitative description of the seeds based on their lateral views, and for this purpose, eight models (geometric figures related to the cardioid) encoded by different equations were obtained that describe the morphology of seeds in their lateral views for diverse species of *Silene* [19,20].

In reference to the dorsal view of seeds, being aware of the importance of morphology in taxonomy, Rohrbach (1869) [18] introduced a classification of *Silene* seeds based on the structure of the back of the seed as flat (dorso plana) or deepened (dorso canaliculata). These two categories correspond, respectively, to convex and non-convex seeds in their dorsal views (see later). Based on electron microscopy images, several reports describe the micromorphology of the dorsal views of *Silene* seeds [21,22,23,24,25,26], but a systematic approach to general shape in geometric terms has not been done. A first approach to the morphology of the dorsal side [20] showed interesting differences between species in this genus, with seeds predominantly convex in species of *S.* subg. *Behenantha*, such as *S. conica*, *S. foetida* Link ex Spreng, *S. latifolia*, and *S. littorea*, and non-convex in the other species. Therefore, we would like to investigate further this aspect. In this work, convexity is referred to a particular feature of the planar image of the dorsal view of the seed. The corresponding region is said to be *convex* when the straight line joining any pair of internal points lies entirely inside the region. This definition concerns overall shape and does not consider surface protuberances or colliculae that often make up a small percentage of the total surface area.

The general objective of this work was to analyze in detail the morphological characteristics of the dorsal views of seeds, and, more specifically, to obtain new geometric models derived from algebraic equations that describe the shape of the dorsal views of representative *Silene* species, allowing shape quantification for their identification and classification. Both the lateral and dorsal views of seeds were analyzed by the comparison with geometric models. Values of *J* index, indicating the percentage of similarity between the seed images and the models, were obtained in different populations of some species. The results contribute to the description of the dorsal view of seeds in *Silene* species and give, for the first time, data on the variation of seed shape in populations and species.

## 2. Results

### 2.1. Variability in Size and Shape in the Dorsal View of Seeds

Two groups of taxa with convex and non-convex seeds were analyzed separately because their respective models are quite different. While convex models form a continuity, and there is not a straight relationship of one model–one species, non-convex models are more species-specific.

Table 1 and Table 2 show the mean values and standard deviations for area (A), perimeter (P), length (L), width (W), aspect ratio (AR), circularity (C), and roundness (R), together with the results of Kruskal–Wallis and post hoc tests to compare the distributions for species of convex and non-convex seeds, respectively.

#### 2.1.1. Variability in Size and Shape in Convex Seeds

The area of convex seeds was comprised between 0.45 (*S. littorea*) and 1.65 mm^2^ (*S. vulgaris*), with differences between species (Table 1). Differences between all species were found except for the groups formed by *S. diclinis* and *S. vulgaris*, on one side, and *S. dioica* and *S. latifolia*, on the other. Identical results were obtained for length, and very similar for perimeter and width. The values of aspect ratio were comprised between 1.43 (*S. latifolia*) and 1.56 (*S. littorea*). The values of roundness were opposed to aspect ratio varying between 0.64 (*S. littorea*) and 0.70 (*S. latifolia*). The values of circularity were comprised between 0.49 (*S. foetida*) and 0.79 (*S. littorea*).

#### 2.1.2. Variability in Size and Shape in Non-Convex Seeds

The area of non-convex seeds was comprised between 0.24 (*S. inaperta* and *S. ramosissima*) and 1.84 mm^2^ (*S. pseudoatocion*) (Table 2). The seeds of *S. pseudoatocion* were the longest and had the highest values of perimeter, width, circularity, and roundness compared to the other species (*p* < 0.05). The values of width, circularity, and roundness were significantly lower in *S. inaperta*. The values of perimeter and length were significantly lower in *S. ramosissima*. Statistically, both species *S. inaperta* and *S. ramosissima* had the lowest area values. The values of aspect ratio were comprised between 1.17 (*S. pseudoatocion*) and 2.24 (*S. inaperta*). The values of roundness showed a trend opposed to the aspect ratio, being comprised between 0.45 (*S. inaperta*) and 0.86 (*S. pseudoatocion*).

### 2.2. New Models Representative of the Dorsal View of Seeds

#### 2.2.1. Models for Convex Seeds

Four new models (DM1 to DM4) were obtained to define and quantify the shape of convex seeds are shown in Figure 1.

The equations corresponding to models DM1 to DM4 (DM stands for Dorsal Model) are as follows: Model DM1 is a superellipse (squared circle) of the following equation:$${\left|\frac{3x}{2}\right|}^{3}+{\left|y\right|}^{3}=1$$

Models DM2 to DM4, as well as DM5–DM8 (see Section 2.2.2), were based on the equation of an ellipse as it was described for the Arecaceae and the Vitaceae [12,14].

DM2 corresponds to the following equation:$$\left(-y-\frac{10\sqrt{40-x^{2}}}{11}+\frac{5}{4\left({\left(\frac{10\;x^{2}}{363}\right)}^{30}+12\right)}\right)\left(-y+\frac{10\sqrt{40-x^{2}}}{11}+\frac{5}{4{\left(\frac{10\;x^{2}}{363}\right)}^{30}+\frac{7}{10}}\right)=0$$

DM3 was obtained from the representation of the following equation:$$\left(\frac{9\;y}{10}-\frac{5\sqrt{40-x^{2}}}{4}+\frac{5}{4\left(\frac{10\;x^{2}}{432}+4\right)}\right)\left(\frac{4\;y}{5}+\frac{5\sqrt{40-x^{2}}}{4}+\frac{5}{6\left({\left(\frac{10\;x^{2}}{432}\right)}^{10}+4\right)}\right)=0$$

Model DM4 was obtained from the representation of the following equation:$$\left(-y-\frac{10\sqrt{40-x^{2}}}{11}+\frac{5}{4\left(\frac{10\;x}{11}+12\right)}\right)\left(-y+\frac{10\sqrt{40-x^{2}}}{11}+\frac{5}{4{\left(\frac{10\;x^{2}}{363}\right)}^{6}+\frac{3}{4}}\right)=0$$

#### 2.2.2. Models for Non-Convex Seeds

Five new models (DM5 to DM9) were obtained to define and quantify the shape of non-convex seeds, as shown in Figure 2.

Model DM5 was obtained from the representation of the following equation:$$\left(y-\frac{9\sqrt{33-x^{4}}}{10}+\frac{40}{11\left(x^{4}+x^{2}+6\right)}\right)\left(y+\frac{\sqrt{33-x^{4}}}{2}-\frac{50}{11\left(x^{4}+x^{2}+6\right)}\right)=0$$

Model DM6 was obtained from the representation of the following equation:$$\left(\frac{9\;y}{10}-\frac{9\sqrt{33-x^{4}}}{10}+\frac{30}{11\left(x^{4}+x^{2}+6\right)}\right)\left(\frac{9\;y}{10}+\frac{\sqrt{33-x^{4}}}{2}-\frac{50}{11\left(x^{4}+x^{2}+6\right)}\right)=0$$

Model DM7 was obtained from the representation of the following equation:$$\left(y-\frac{6\sqrt{33-x^{2}}}{5}+\frac{50}{11\left(x^{4}+x^{2}+6\right)}\right)\left(y+\frac{6\sqrt{33-x^{2}}}{5}-\frac{50}{11\left(x^{4}+x^{2}+6\right)}\right)=0$$

Model DM8 resulted from the following equation:$$\left(3y-6\sqrt{33-x^{2}}+\frac{250}{11\left(x^{4}+x^{2}+6\right)}\right)\left(3y+6\sqrt{33-x^{2}}-\frac{250}{11\left(x^{4}+x^{2}+6\right)}\right)=0$$

Model DM9 was obtained from the representation in polar coordinates of the following equation:$$\rho ={\left(\cos^{20}\left(\theta \right)+4\;\sin^{2}(\theta )\right)}^{-2/3}$$

### 2.3. Application of the Models to Seed Shape Quantification in the Dorsal Views of Silene Species

The following two sub-sections describe the application of new models to the analysis of seed shape in species with convex and non-convex seeds, respectively.

The percent of similarity between the image of a seed and the model is given by *J* index.

#### 2.3.1. Quantification with Models for Convex Seeds

In the lateral views, the models giving the best scores confirmed the results reported before [19,20]: LM1 for *S. littorea*; LM2 for *S. dioica*, *S. foetida*, *S. latifolia*, and *S. vulgaris*; LM3 and LM6 for *S. gallica*; and LM4 for *S. diclinis* (not shown).

Table 3 contains the results of Kruskal–Wallis and post hoc tests for values of *J* index in the comparisons between species of convex seeds with dorsal models DM1 to DM4.

The four first new models (DM1 to DM4) resembled the shape of convex seeds, though not in a very significant way. Values of *J* index superior to 90 were obtained with DM1 in the seeds of *S. diclinis, S. gallica*, and *S. latifolia*; with DM2 in *S. foetida, S. gallica, S. latifolia*, and *S. vulgaris*; with DM3 in *S. diclinis* and *S. dioica*; and, finally, with DM4 in *S. littorea*. Differences between species were found for all the models. With model DM1, differences were found between *S. diclinis* and the other species except *S. gallica* (*p* < 0.05). With model DM2, *J* index was higher in *S. gallica* and *S. latifolia* than in the other species (*p* < 0.05). *S. foetida* and *S. gallica* gave lower values with model DM3 than the other species. *S. littorea* gave higher values than the other species except *S. vulgaris* with model DM4.

A clustering analysis was made to refine these results. According to the heatmap in Figure 3, *S. latifolia*, *S. diclinis*, and *S. littorea* showed high values for DM2, DM1, and DM4, correspondingly. *S. foetida* showed average low levels, the highest value being for DM2. None of the species showed significant affinity with DM3. Clustering showed high similarity between DM1 and DM2 as well as for DM3 and DM4.

Figure 4 shows the image of 20 seeds of *S. diclinis* superimposed, their average dorsal silhouette, together with the model giving the best adjustment (DM1), and two representative seeds of this species.

Figure 5 shows 20 seeds of *S. gallica* superimposed, their average dorsal silhouette, the model DM2 (model with the best *J* index adjustment), and two representative seeds of this species.

Figure 6 shows 20 seeds of *S. latifolia* superimposed, their average dorsal silhouette, the model DM2 (model with the best *J* index adjustment), and two representative seeds of this species.

Figure 7 presents 20 seeds of *S. dioica* superimposed, their average dorsal silhouette, the model DM3 (model with the best *J* index adjustment), and two representative seeds of this species.

Figure 8 presents 20 seeds superimposed of *S. littorea*, their average dorsal silhouette, the model DM4 (model with the best *J* index adjustment), and two representative seeds of this species.

#### 2.3.2. Quantification with Models for Non-Convex Seeds

In the lateral views, the models giving the best scores confirmed the results reported before with LM1 for *S. conica* and LM7 for *S. coutinhoi* [19,20]. In addition, values of *J* index of 90.1 were obtained with LM3 for *S. pseudoatocion*, of 91.9 with LM7 for *S. ramosissima*, and 91.0 with LM8 for *S. inaperta*.

Table 4 contains the results for species of non-convex seeds with lateral models LM1, LM3, LM7, and LM8 [19,20], and with dorsal models DM5 to DM9. It shows the mean and standard deviation values of the *J* index obtained with the different models in the species and populations of non-convex seeds. In contrast with the results obtained for convex seeds, the models for non-convex seeds were highly specific for the species tested. Therefore, the model DM5 adjusted well to the seeds of *S. conica*, the model DM6 to *S. coutinhoi*, DM7 to *S. pseudoatocion*, DM8 to *S. ramosissima*, and DM9 to *S. inaperta*. The models are quite specific for their respective species, and the values of *J* index obtained for each model in species other than their own were below 80. The results of a clustering analysis are shown in the heatmap in Figure 9 and illustrate the relationship between seed shape in each species and their respective models.

Figure 10, Figure 11, Figure 12, Figure 13 and Figure 14 show the seed images corresponding to species characterized by non-convex seeds (Table 4) with their respective models: *S. conica* with DM5 (Figure 10), *S. coutinhoi* with DM6 (Figure 11), *S. pseudoatocion* with DM7 (Figure 12), *S. ramosissima* with DM8 (Figure 13), and *S. inaperta* with DM9 (Figure 14). The figures contain the image of 20 seeds superimposed, average dorsal silhouette of each population, together with the corresponding model, and two or three seeds representative of each population.

## 3. Discussion

The description and quantification of seed shape by comparison with geometric models were applied here for the third time to species of *Silene*. Based on this method, seed shape was quantified by the percent similarity between seed images and the corresponding geometric model (*J* index). This method allows the comparison of shape measurements between species and their populations. The first report included 21 species, 11 of them in *S.* subg. *Behenanatha* and 10 in *S.* subg. *Silene* with different populations in some of them [19]. The second approach studied 21 species of which five had already been included in the previous work (*S. conica*, *S. latifolia*, and *S. vulgaris* in *S.* subg. *Behenantha* and *S. gallica* and *S. mellifera* in *S.* subg. *Silene*), while the remaining 16 species were studied for the first time with this technique [20]. In summary, results for a total of 37 different species were published before, all of them related with the lateral views of the seeds. This work presents results with three new species and several new populations of others previously reported, thus reaching a total of 40 species whose shape has been investigated so far by the comparison with geometric models. A total of eight models for the lateral views were described for quantification of the lateral view in *Silene* species, with values of *J* index superior to 90 in many of them [19,20].

Model 1 (LM1), the cardioid, resulted in maximum scores with seeds of *S. noctiflora* L., and *J* index with this model was superior to 90 in many species of *Silene*, as reported by Martín Gómez et al. [19] and Juan et al. [20]. In the former work, values superior to 90 were found in the 11 species tested of *S.* subg. *Behenantha* (*S. acutifolia* Link ex Rohrb., *S. conica* L., *S. diclinis* (Lag.) M. Laínz, *S. dioica* (L.) Clairv., *S. latifolia* Poir., *S. noctiflora* L., *S. pendula* L., *S. uniflora* Roth., *S. viscosa* Pers., *S. vulgaris* (Moench) Garcke, and *S. zawadskii* Fenzl) as well as in five species belonging to *S.* subg. *Silene* (*S. gallica* L., *S. italica* (L.) Pers., *S. nutans* L., *S. otites* Sm., *S. saxifraga* L.). In the second report [20], values of *J* index superior to 90 with the cardioid were obtained for *S. conica*, *S. littorea* Brot., *S. portensis* L., and *S. vivianii* Steud. In both works, the mean values of *J* index with the cardioid were superior in *S.* subg. *Behenantha* than in *S*. subg. *Silene*. The other models tested in addition to the cardioid differed mainly in the hilum region, were less prominent in models LM2 and LM4, and were more pronounced in models LM3, LM5, and LM6. The obtained models LM7 and LM8 had other properties including a slight elongation and a certain asymmetry in the latter. The species *S. diclinis*, *S. latifolia*, and *S. littorea* had higher values with models LM2 and LM4 than with the cardioid, while *S. gallica* had the highest values with LM3 [19]. Finally, LM2 was associated with convex type seeds in the dorsal views, while the other models were more frequent with non-convex seeds [20].

The affinity with the different models was also tested on the average silhouette, an image resulting from the composition of a shared area of twenty representative seeds of each species [20]. This analysis suggested a relationship between the models adjusting to the lateral views and the shape of the dorsal view. Species with a better adjustment to models LM2 and LM4 in the lateral views had convex seeds in the dorsal views, while those species characterized by other lateral models (LM3, LM5, LM7, and LM8) showed their dorsal views as non-convex [20]. In addition, the analysis of the seed dimensions of the dorsal views revealed three groups, well distinguished on the basis of their different roundness and aspect ratio [20].

To further gain insight in the relationship between seed shape in the lateral and dorsal views, we applied in this work a series of new models designed to describe and quantify the dorsal views of representative *Silene* species of both convex and non-convex types (DM1 to DM4 and DM5 to DM9, respectively). Our results indicated less variation in shape than in size, and still less variation for *J* index than for other general shape measurements (aspect ratio, circularity, roundness). This is similar to those results obtained for the analysis of seed shape in genera of the Cactaceae [13].

The models DM1 to DM4 adjusted well to the dorsal views of convex seeds with *J* index values in some instances higher than 90. For these convex seeds it was not possible to ascribe a specific model to each species, but the combination of values with the different models gave some definition for the dorsal view in five species. Therefore, *S. diclinis* seeds adjusted well to DM1, *S. gallica* to DM1 and DM2, *S. latifolia* to DM2, *S. dioica* and *S. littorea* to DM3 and DM4, and *S. vulgaris* to DM3. In contrast, *S. foetida* remained less defined by the models. The combination of *J* index values obtained for the lateral and dorsal view gave a good definition of the shape of some of the species; thus, *S. diclinis* could be distinguished from *S. latifolia* by higher values of *J* index with model LM4 and lower values with model LM2 and DM2.

Nevertheless, the relationship between species and models was notably better defined in the group of non-convex seeds, since each studied species was clearly related to a specific model. The dorsal shape of *S. conica* seeds was well defined by DM5, with a *J* index value of 91.3. For *S. coutinhoi*, their seeds were well described by DM6 with a *J* index value of 90.7, and the dorsal views of *S. inaperta*, *S. pseudoatocion*, and *S. ramosissima* corresponded to models DM9, DM7, and DM8, respectively. These results are quite remarkable, since the specificity of the geometrical models might be considered as an additional morphological tool to support the differentiation of some complex groups of this genus.

The description and quantification of seed shape based on geometric models revealed a certain level of differentiation among species by the shape of their seeds. Although the number of studied species of the genus *Silene* was low, some preliminary tendencies were noted based on the relationships between the presence of convex/non-convex seeds and lifespan and taxonomical support. The majority of the studied annual species was likely to have non-convex seeds, whereas the perennial species were mostly characterized by convex seeds. Moreover, the convex seeds were mostly observed for the different species of *S*. subg. *Behenantha*, but more studies are needed to support these preliminary data and to test the potential use of these geometrical models for the identification of the current taxonomical treatment on sections. Further studies are needed to show the adjustment for other species to the models here described, and particularly for the convex models, and to increase the number of *Silene* species to check the initial observations described here.

Fourier analysis has been applied to seed shape quantification in a number of plant species [27,28,29,30], allowing distances between shapes to be quantified, as well as defining the average silhouette for each shape type. It permits one to analyze the variation both within and between morphological types, independently from the similarity to any geometric figure. In contrast, the experimental setup applied in this work compares bi-dimensional seed shape with a figure of reference. In both cases, the results are not only numerical and statistical but analytical, while the second one provides additionally geometric models. As a consequence, the method here applied presents a supplementary approach to Fourier analysis, providing the similarity of a seed image with a given geometric figure. In this way, the model gives a geometric reference, with a corresponding algebraic equation, to which the description of seed shape may be associated. In an ideal case, Fourier analysis could be done first, and the results complemented with an analytical approach wherein the infinitive possible shapes are constrained to just a few specific ones, identified as “paradigmatic” and represented by geometric models.

The species described in this work include among others also some of the commonest in Europe and the Mediterranean region, and also some of the best characterized genetically. Whole-genome draft sequence data are so far available from the widespread dioecious species *Silene latifolia* [31,32,33] and from its relatively close widespread hermaphrodite relative—*S. noctiflora* [34]. The genus *Silene* serves as an important model system in ecology and evolution [35]. The studied topics include, e.g., evolution of sex determining systems, epigenetic aspects of sex determination, male sterility, and evolution of organellar genomes. *S. noctiflora* and *S. conica* have some of the largest known plant mitochondrial genomes of 7 Mb and 11 Mb, respectively, in contrast with the small size for the mitochondrial genome of *S. latifolia*, estimated at 0.25 Mb [36]. In the genus *Silene*, multiple transitions occurred in the breeding systems and there occurred also changes connected with accommodation to different environments (e.g., adaptation of *S. ramosissima*, *S. nicaeensis*, and *S. succulenta* to the dune environment [37,38]). It may be interesting to find out if there is some correlation between some of these changes and the morphological changes described by the use of models. Future progress in whole genome sequencing and construction of dense genetic maps can enable one to apply model based evaluation of morphological traits in mapping of corresponding QTLs and the identification of the genes involved.

## 4. Materials and Methods

### 4.1. Seeds Used in This Study

Table 5 presents the populations used in this work. It includes 26 populations belonging to 12 species (3 populations of *S. conica*, 1 of *S. coutinhoi*, 3 of *S. diclinis*, 3 of *S. dioica*, 1 of *S. foetida*, 2 of *S. gallica*, 3 of *S. inaperta*, 6 of *S. latifolia*, 1 of *S. littorea*, 1 of *S. pseudoatocion* 1 of *S. ramosissima*, and 1 of *S. vulgaris*). With the only exception of *S. latifolia* 06, the collected seeds came from natural populations, each with its own particular environmental conditions. The species were chosen to investigate differences in shape between both subgenera, *Silene* and *Behenantha*, in support of results reported before and to explore in detail these differences [18,20].

### 4.2. Seed Images

Photographs of the seeds collected in Spain and Portugal were taken in Salamanca with a Nikon Stereomicroscope Model SMZ1500 (Nikon, Tokyo, Japan) equipped with a Nikon DS-Fi1 camera of 5.24 megapixels, and all the other seeds were photographed in Brno with an Olympus SZX9 microscope (Olympus, Japan) coupled to an Olympus OM-D E-M10 II camera. Composed images containing 20 seeds per accession were prepared with Corel Photo Paint for the lateral and dorsal views of the seeds. The images are stored at https://zenodo.org/record/5997053#.YgE1eOrMKM8 (accessed on 15 March 2022).

### 4.3. General Description of Size and Shape

Measurements of area (A), perimeter (P), length of the major axis (L), width (W), aspect ratio (AR is the ratio L/W), circularity (C), and roundness (R) were obtained for the dorsal views of seeds of each population with ImageJ [42]. A ruler was included in the photographs for the conversion of pixel units to length or surface units (mm or mm^2^). The circularity index and roundness were measured as described [43]. Circularity is the ratio (4π × A)/P2, while roundness is (4 × A)/πL2; in consequence, irregularities of the seed surface that increase the perimeter reduce the values of circularity, leaving roundness unaffected.

### 4.4. Obtaining the Average Silhouette for a Group of Seeds

The average silhouette is a representative image of seed shape for each group of seeds. A total of 20–30 seeds was used for each population. The silhouette was obtained in Corel Photo Paint, by the protocol described [44] (a detailed video is available at Zenodo: https://zenodo.org/record/4478344#.YBPOguhKiM8, accessed on 2 September 2021). The layers containing the seeds were superimposed, and the opacity was given a value of 3 in all layers. All the layers were combined, and the brightness was adjusted to a minimum value. From this image, we were interested in the inner region representing the area where most of the seeds coincided, which is the darkest area. To select it, we used the magic wand tool, and with tolerance equal to 10, this selection was copied and pasted as a new layer.

### 4.5. Seed Shape Quantification and Testing of the Models: J Index

*J* index indicates the similarity between a bi-dimensional image of a seed and the corresponding model. *J* index is the percent of area that is shared by both images once they are superimposed for maximum similarity. It is quantified as:J=S/T×100,
where S is the area shared between the seed and the model, and T is the total area occupied by both images. *J* index has a maximum value of 100, corresponding to the cases where the geometric model and the seed image areas coincide. High scores indicate similarity of a seed image with a given model, meaning that the model provides a precise definition of seed shape for a particular species. A good adjustment to the model is considered when *J* index values are greater than 90 [19]. Figure 15 shows an example of the images used for J index quantification with the average silhouette obtained for *S. coutinhoi*.

Once a model was found adjusting to seeds of one species, the values of *J* index were obtained for all the populations of this species as well as for a number of populations from other species.

### 4.6. Statistical Analysis

The raw data are available at: https://zenodo.org/record/6276242#.YhiliJaCGUk (accessed on 15 March 2022). The box-plot representations corresponding to the populations under study are available at: https://zenodo.org/record/6298757#.Yhn4TujMKM8 (accessed on 15 March 2022). The mean, minimum, and maximum values and the standard deviation were obtained for all the measurements indicated above (A, P, L, W, AR, C, and R) as well as for *J* index with the different models. Statistics was done on IBM SPSS statistics v28 (SPSS 2021) and R software v. 4.1.2 [45]. As some of the populations did not follow a normal distribution, non-parametric tests were applied for the comparison of populations. Kruskal–Wallis tests were done in the cases involving three or more populations (Table 1, Table 2 and Table 3), followed by stepwise stepdown comparisons by the ad hoc procedure developed by Campbell and Skillings [46]; *p* values inferior to 0.05 were considered significant. The coefficient of variation was calculated as CVtrait = standard deviationtrait/meantrait × 100 [47].

## 5. Conclusions

Nine new geometric models were presented to describe and quantify the dorsal views of *Silene* seeds. Four of them (DM1 to DM4) were applied to convex seeds of the species *S. diclinis*, *S. dioica*, *S. gallica*, *S. latifolia*, *S. foetida*, and *S. vulgaris*, while the remaining five models (DM5 to DM9) were designed to adjust the shape of non-convex seeds in *S. conica, S. coutinhoi, S. gallica, S. inaperta, S. pseudoatocion*, and *S. ramosissima.* There is more specificity in the models described for the non-convex seeds in the sense that seeds of a species adjust to their corresponding model better than the seeds of other species. For example, the models DM6, DM7, DM8, and DM9 adapt well to the seeds of *S. coutinhoi, S. pseudoatocion, S. inaperta*, and *S. ramosissima*, respectively. The combination of models for the lateral and dorsal views can give a good definition of seed shape in some species of *Silene*.

Seed shape has been quantified in diverse populations for some species, and the comparison of the coefficient of variation for size and shape measurements allows us to conclude that shape varies less than size in populations of *Silene* seeds.

## Figures and Tables

**Figure 1 plants-11-00958-f001:**
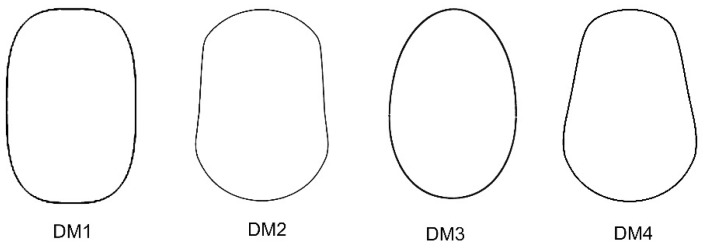
New models for the dorsal views of convex seeds in *Silene* species. DM stands for Dorsal Model.

**Figure 2 plants-11-00958-f002:**
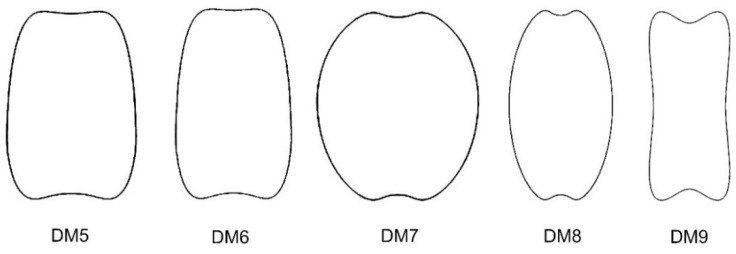
New models for the dorsal views of non-convex seeds in *Silene* species.

**Figure 3 plants-11-00958-f003:**
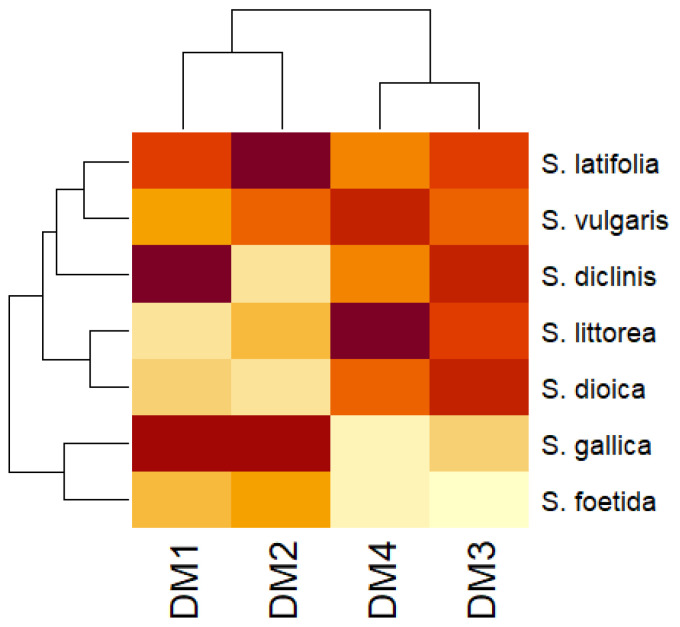
Clustering analysis for species of convex seeds with models DM1 to DM4 according to Euclidean distance. Dark red stands for high values and light yellow for low values.

**Figure 4 plants-11-00958-f004:**
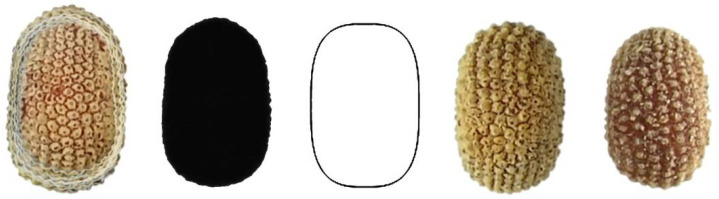
Left to right: Dorsal view of twenty seeds of *S. diclinis* superimposed; the corresponding average silhouette; model DM1, and two representative seeds of *S. diclinis*.

**Figure 5 plants-11-00958-f005:**
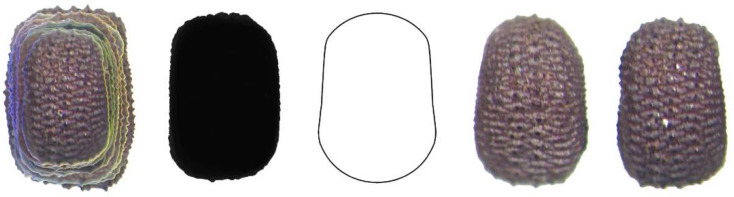
Left to right: Dorsal view of twenty seeds of *S. gallica* superimposed; the corresponding average silhouette; model DM2, and two representative seeds of this species.

**Figure 6 plants-11-00958-f006:**
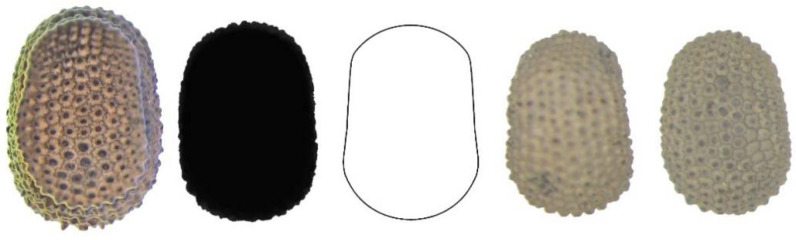
Left to right: Dorsal view of twenty seeds of *S. latifolia* superimposed; the corresponding average silhouette; model DM2, and two representative seeds of *S. latifolia*.

**Figure 7 plants-11-00958-f007:**

Left to right: Dorsal view of twenty seeds of *S. dioica* superimposed; the corresponding average silhouette; the model DM3, and two representative seeds of *S. dioica*.

**Figure 8 plants-11-00958-f008:**
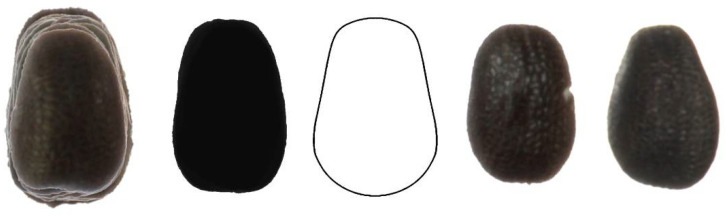
Left to right: Dorsal view of twenty seeds of *S. littorea* superimposed; the corresponding average silhouette; model DM4, and two representative seeds of this species.

**Figure 9 plants-11-00958-f009:**
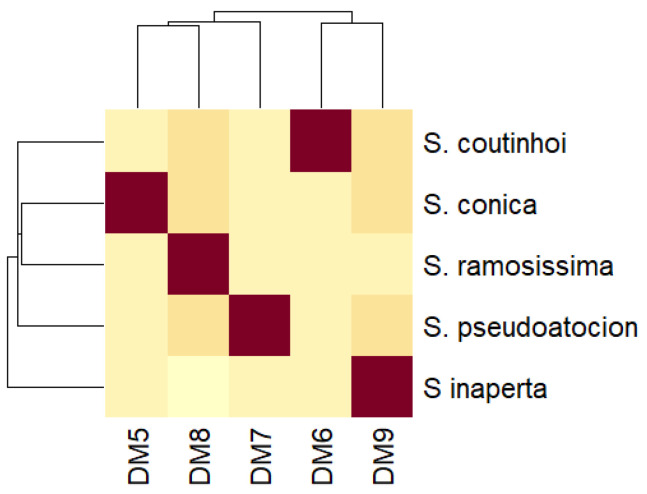
Clustering analysis for species of non-convex seeds with models DM5 to DM9 according to Euclidean distance. Dark red stands for high values and light yellow for low values.

**Figure 10 plants-11-00958-f010:**
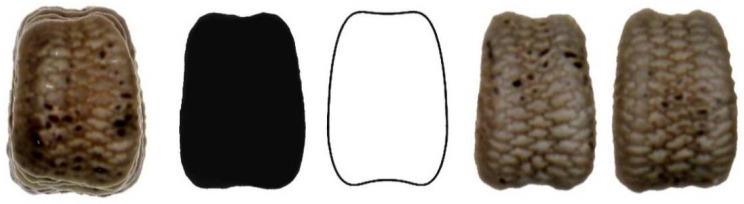
Left to right: Dorsal view of twenty seeds of *S. conica* superimposed; the corresponding average silhouette; the model DM5, and two representative seeds of *S. conica* 01.

**Figure 11 plants-11-00958-f011:**
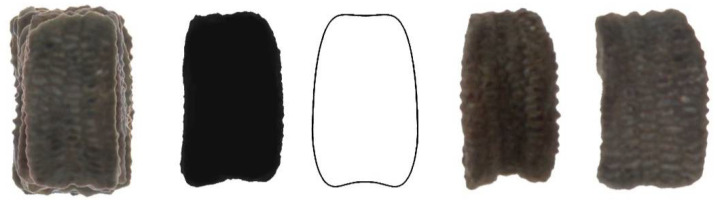
Left to right: Dorsal view of twenty seeds of *S. coutinhoi* superimposed; the corresponding average silhouette; the model DM6, and two representative seeds of *S. coutinhoi*.

**Figure 12 plants-11-00958-f012:**
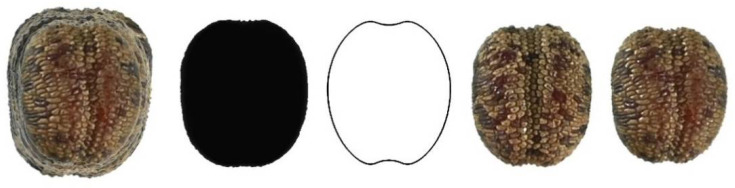
Left to right: Dorsal view of twenty seeds of *S. pseudoatocion* superimposed; the corresponding average silhouette; the model DM7, and two representative seeds of *S. pseudoatocion*.

**Figure 13 plants-11-00958-f013:**
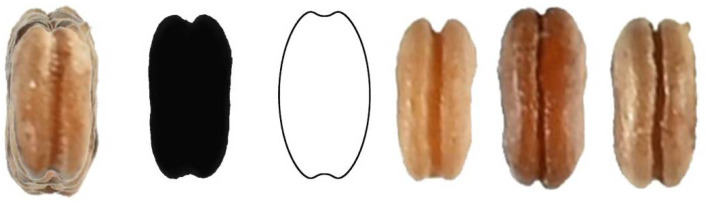
Left to right: Dorsal view of twenty seeds of *S. ramosissima* superimposed; the corresponding average silhouette; the model DM8, and three representative seeds of *S. ramosissima*.

**Figure 14 plants-11-00958-f014:**
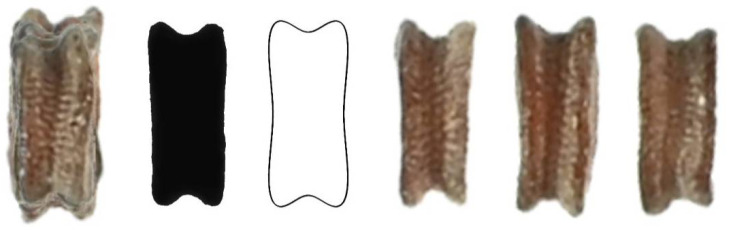
Left to right: Dorsal view of twenty seeds of *S. inaperta* superimposed; the corresponding average silhouette; model DM9, and three representative seeds of *S. inaperta*.

**Figure 15 plants-11-00958-f015:**
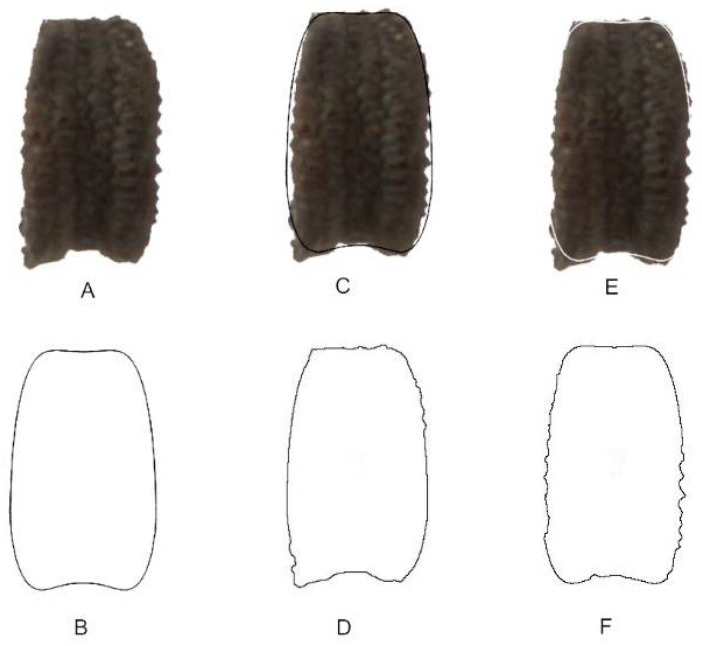
Graphic representation of the method used to calculate *J* index: (**A**) dorsal view of a seed from *S. coutinhoi*; (**B**) the dorsal model DM6; (**C**) images in (**A**,**B**) superimposed searching a maximum coincidence; (**D**) interpretation of image (**C**) after adjustments (8 bit and color threshold) with ImageJ; (**E**) same as in (**C**) but with the model in white, obtained by giving maximum brightness to the image of the model; (**F**) interpretation of image (**E**) after adjustments (8 bit and color threshold) with ImageJ. The area measured in (**D**) with ImageJ is the total area (T), while the area measured in (**F**) corresponds to the shared area (S) between the seed and the model. The values for the estimated areas are 39,998 pixels for the shared (S) and 44,223 pixels for the total area (T), respectively. The *J* index of this example is equal to 90.4.

**Table 1 plants-11-00958-t001:** Results of Kruskal–Wallis and post hoc tests for the species with convex seeds. Mean values and standard deviations (given in parentheses) are indicated for area (A), perimeter (P), length (L), width (W), aspect ratio (AR), circularity (C), and roundness (R). Values marked with the same superscript letter in each column correspond to populations that do not differ significantly at *p* < 0.05 (Campbell and Skillings test). N indicates the number of seeds analyzed.

Species	N	A	P	L	W	AR	C	R
*S. diclinis*	60	1.64 ^e^ (0.20)	5.44 ^d^ (0.47)	1.79 ^e^ (0.11)	1.16 ^f^ (0.08)	1.54 ^d^ (0.07)	0.70 ^cd^ (0.06)	0.65 ^a^ (0.03)
*S. dioica*	60	0.99 ^d^ (0.16)	4.40 ^c^ (0.57)	1.36 ^d^ (0.12)	0.92 ^d^ (0.07)	1.47 ^b^ (0.09)	0.65 ^b^ (0.07)	0.68 ^c^ (0.04)
*S. foetida*	19	0.77 ^c^ (0.06)	4.44 ^c^ (0.30)	1.21 ^c^ (0.06)	0.81 ^c^ (0.05)	1.47 ^bc^ (0.07)	0.49 ^a^ (0.04)	0.68 ^bc^ (0.04)
*S. gallica*	40	0.69 ^b^ (0.11)	3.48 ^b^ (0.35)	1.15 ^b^ (0.10)	0.77 ^b^ (0.06)	1.48 ^bc^ (0.05)	0.71 ^d^ (0.03)	0.68 ^bc^ (0.03)
*S. latifolia*	120	1.01 ^d^ (0.16)	4.30 ^c^ (0.38)	1.35 ^d^ (0.10)	0.94 ^e^ (0.09)	1.43 ^a^ (0.07)	0.68 ^c^ (0.06)	0.70 ^d^ (0.04)
*S. littorea*	19	0.45 ^a^ (0.04)	2.67 ^a^ (0.10)	0.94 ^a^ (0.06)	0.61 ^a^ (0.03)	1.56 ^d^ (0.09)	0.79 ^e^ (0.02)	0.64 ^a^ (0.03)
*S. vulgaris*	16	1.65 ^e^ (0.15)	5.70 ^e^ (0.31)	1.78 ^e^ (0.08)	1.18 ^f^ (0.06)	1.51 ^cd^ (0.05)	0.64 ^b^ (0.03)	0.66 ^ab^ (0.02)

**Table 2 plants-11-00958-t002:** Results of Kruskal–Wallis and post hoc tests for the species with non-convex seeds. Mean values and standard deviations (given in parentheses) are indicated for area (A), perimeter (P), length (L), width (W), aspect ratio (AR), circularity (C), and roundness (R). Values marked with the same superscript letter in each column correspond to populations that do not differ significantly at *p* < 0.05 (Campbell and Skillings test). N indicates the number of seeds analyzed.

Species	N	A	P	L	W	AR	C	R
*S. conica*	57	0.49 ^b^ (006)	2.92 ^c^ (0.15)	0.95 ^c^ (0.07)	0.66 ^c^ (0.04)	1.43 ^b^ (0.06)	0.72 ^c^ (0.05)	0.70 ^d^ (0.03)
*S. coutinhoi*	16	0.55 ^c^ (0.05)	3.27 ^d^ (0.12)	1.09 ^d^ (0.03)	0.65 ^c^ (0.05)	1.69 ^c^ (0.14)	0.65 ^b^ (0.03)	0.59 ^c^ (0.05)
*S inaperta*	70	0.24 ^a^ (0.03)	2.34 ^b^ (0.11)	0.83 ^b^ (0.05)	0.37 ^a^ (0.03)	2.24 ^e^ (0.19)	0.56 ^a^ (0.05)	0.45 ^a^ (0.04)
*S. pseudoatocion*	20	1.84 ^d^ (0.29)	5.31 ^e^ (0.41)	1.65 ^e^ (0.14)	1.41 ^d^ (0.12)	1.17 ^a^ (0.05)	0.82 ^d^ (0.02)	0.86 ^e^ (0.03)
*S. ramosissima*	20	0.24 ^a^ (0.03)	2.10 ^a^ (0.13)	0.77 ^a^ (0.06)	0.39 ^b^ (0.02)	2.00 ^d^ (0.09)	0.67 ^b^ (0.04)	0.50 ^b^ (0.02)

**Table 3 plants-11-00958-t003:** Results of Kruskal–Wallis and post hoc tests for the species with convex seeds. Mean values and standard deviations (given in parentheses) for *J* index values with dorsal models DM1 to DM4. Values marked with the same superscript letter in each column correspond to populations that do not differ significantly at *p* < 0.05 (Campbell and Skillings test). N indicates the number of seeds analyzed.

Species	N	*J*(DM1)	*J*(DM2)	*J*(DM3)	*J*(DM4)
*S. diclinis*	72	90.9 ^c^ (1.20)	89.4 ^ab^ (3.07)	90.0 ^b^ (1.67)	89.0 ^b^ (1.68)
*S. dioica*	60	88.8 ^a^ (1.59)	89.3 ^a^ (1.53)	90.1 ^b^ (1.96)	89.3 ^b^ (1.61)
*S. foetida*	19	89.1 ^a^ (1.67)	90.2 ^ab^ (1.40)	87.2 ^a^ (1.78)	87.6 ^a^ (1.77)
*S. gallica*	40	90.7 ^bc^ (1.47)	91.4 ^c^ (1.17)	88.1 ^a^ (1.84)	87.6 ^a^ (1.65)
*S. latifolia*	120	90.2 ^b^ (1.96)	91.7 ^c^ (1.63)	89.7 ^b^ (2.55)	89.0 ^b^ (2.06)
*S. littorea*	19	88.6 ^a^ (2.26)	89.8 ^ab^ (2.51)	89.8 ^b^ (2.31)	90.6 ^c^ (2.07)
*S. vulgaris*	16	89.4 ^ab^ (1.96)	90.7 ^b^ (1.31)	89.3 ^b^ (1.92)	89.8 ^bc^ (1.16)

**Table 4 plants-11-00958-t004:** Mean values and standard deviations (given in parentheses) for *J* index values with lateral (LM) LM1, LM3, LM7, and LM8, and dorsal (DM) models DM5, DM6, DM7, DM8, and DM9. N indicates the number of seeds analyzed.

Species	N	*J*(LM)	*J*(DM)
*S. conica*	57	90.1 (LM1) (1.89)	91.3 (DM5) (1.91)
*S. coutinhoi*	16	88.5 (LM7) (2.92)	90.7 (DM6) (2.14)
*S inaperta*	70	91.0 (LM8) (1.65)	85.6 (DM9) (2.83)
*S. pseudoatocion*	20	90.1 (LM3) (1.26)	93.1(DM7) (0.83)
*S. ramosissima*	20	91.9 (LM7) (2.65)	87.9(DM8)(1.58)

**Table 5 plants-11-00958-t005:** Seed species and populations used in this study. The populations of each species are labelled according to a code (Lab. Code) corresponding to the different geographical locations (origin). The life span and annual and perennial plants is indicated according to Talavera [39] and Morton [40]. The ascription of each species to subgenera and sections is taken from [41].

Species	Lab. Code	Origin	Life Span	Subgenera/Section
*S. conica* L.	*S. conica* 01	Villena, Alicante (Spain)	Annual	*S*. subg. *Behenantha* sect. *Conoimorpha*
*S. conica* L.	*S. conica* 02	Botanic Garden, Berlin (Germany);	Annual	*S*. subg. *Behenantha* sect. *Conoimorpha*
*S. conica* L.	*S. conica* 03	Humboldt-Universitat Berlin (Germany) 755/219	Annual	*S*. subg. *Behenantha* sect. *Conoimorpha*
*S. coutinhoi* Rothm. and P.Silva	*S. coutinhoi*	Larouco, Ourense (Spain)	Perennial	*S*. subg. *Silene* sect. *Siphonomorpha*
*S. diclinis* (Lag.) M.Laínz	*S. diclinis* 01	Unknown	Perennial	*S.* subg. *Behenantha* sect. *Melandrium*
*S. diclinis* (Lag.) M.Laínz	*S. diclinis* 02	Unknown	Perennial	*S.* subg. *Behenantha* sect. *Melandrium*
*S. diclinis* (Lag.) M.Laínz	*S. diclinis* 03	Unknown	Perennial	*S.* subg. *Behenantha* sect. *Melandrium*
*S. dioica* (L.) Clairv.	*S. dioica* 01	Botanic Garden, Berlin (Germany)	Perennial	*S.* subg. *Behenantha* sect. *Melandrium*
*S. dioica* (L.) Clairv.	*S. dioica* 02	Botanic Garden, Rostock (Germany) F1455_200	Perennial	*S.* subg. *Behenantha* sect. *Melandrium*
*S. dioica* (L.) Clairv.	*S. dioica* 03	Orto Botanico Friulano, Udine (Italy)	Perennial	*S.* subg. *Behenantha* sect. *Melandrium*
*S. foetida* Link ex Spreng.	*S. foetida*	Muiños, Ourense (Spain)	Annual	*S*. subg. *Behenantha* sect. *Acutifoliae*
*S. gallica* L.	*S. gallica* 01	Unknown	Annual	*S*. subg. *Silene* sect. *Silene*
*S. gallica* L.	*S. gallica* 02	St. Gallen Botanical Garden (Switzerland)	Annual	*S*. subg. *Silene* sect. *Silene*
*S. inaperta* L.	*S. inaperta* 01	Unknown	Annual	*S.* subg. *Silene* sect. *Muscipula*
*S. inaperta* L.	*S. inaperta* 02	Elda, Alicante (Spain)	Annual	*S.* subg. *Silene* sect. *Muscipula*
*S. inaperta* L.	*S. inaperta* 03	Petrer, Alicante (Spain)	Annual	*S.* subg. *Silene* sect. *Muscipula*
*S. latifolia* Poir.	*S. latifolia* 01	Pego, Alicante (Spain)	Annual/Perennial	*S*. subg. *Behenantha* sect. *Melandrium*
*S. latifolia* Poir.	*S. latifolia* 02	Brno, neighborhood of Bystrc, South Moravia (Czech Republic). 2007.	Annual/Perennial	*S*. subg. *Behenantha* sect. *Melandrium*
*S. latifolia* Poir.	*S. latifolia* 03	Brno, neighborhood of Bystrc, South Moravia (Czech Republic). 2012.	Annual/Perennial	*S*. subg. *Behenantha* sect. *Melandrium*
*S. latifolia* Poir.	*S. latifolia* 04	Larzac, Dordogne (France).	Annual/Perennial	*S*. subg. *Behenantha* sect. *Melandrium*
*S. latifolia* Poir.	*S. latifolia* 05	Locality of Liblice, district of Mělník, Central Bohemia (Czech Republic).	Annual/Perennial	*S*. subg. *Behenantha* sect. *Melandrium*
*S. latifolia* Poir.	*S. latifolia* 06	Internal laboratory material Institute of Biophysics Academy of Sciences of the Czech Republic. Selfcross (17 generations)	Annual/Perennial	*S*. subg. *Behenantha* sect. *Melandrium*
*S. littorea* Brot.	*S. littorea*	Melides, Baixo Alentejo (Portugal)	Annual	*S*. subg. *Behenantha* sect. *Psammophilae*
*S. pseudoatocion* Desf.	*S. pseudoatocion*	Ibi, Alicante (Spain)	Annual	*S.* subg. *Silene* sect. *Siphonomorpha*
*S. ramosissima* Desf.	*S. ramosissima*	Oliva, Valencia (Spain)	Annual	*S*. subg. *Silene* sect. *Silene*
*S. vulgaris* L.	*S. vulgaris*	Elda, Alicante (Spain)	Perennial	*S*. subg. *Behenantha* sect. *Behenantha*

## Data Availability

The raw data are available at: https://zenodo.org/record/6276242#.YhiliJaCGUk (accessed on 15 March 2022). The box-plot representations corresponding to the populations under study are available at: https://zenodo.org/record/6298757#.Yhn4TujMKM8 (accessed on 15 March 2022). The Mathematica code for Geometric Models is given in: https://zenodo.org/record/5997355#.YgE4g urMKM8 (accessed on 15 March 2022).

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
