# Peer review of "New Geometric Models for Shape Quantification of the Dorsal View in Seeds of Silene Species"

_plants, 2022, doi:10.3390/plants11070958_

Round 1
Reviewer 1 Report
In the manuscript entitled: "New geometric models for shape quantification of the dorsal view in seeds of Silene species" (Number: plants-1611033), Authors provide a very detailed seed classification based on metric data, not only just descriptive approach. This approach represents a good direction because it allows readers to precisely define observed phenomenon, which is not an easy task. The characteristics of Silene seeds are particularly difficult to describe, due to their unique kidney shape. In the presented work, the authors performed a lot of detailed analyzes of the seed shape not only in various Silene species but also within the population, focusing on the dorsal view of seeds in the studied species.
In my opinion in the section “Materials and Methods” there is no information about the climatic conditions in which plants grew and formed seeds taken for the tests. Such information will allow to take into account environmental variability, which is very important in the case of plants.
Besides, maybe it would be worth mentioning in the summary whether such models can also be used in the case of other plant species whose seeds have a kidney shape.
Once again, I think that it is interesting approach, worthwhile to be presented, but a few above minor points should be specified.
Author Response
Dear Reviewer 1:
Thank you for your commentaries that have contributed to improve the quality of the article.
Please find below our responses. In black are your commentaries; in red, our responses and corrections made. Changes have been made according to your instructions as indicated below.
In the manuscript entitled: "New geometric models for shape quantification of the dorsal view in seeds of Silene species" (Number: plants-1611033), Authors provide a very detailed seed classification based on metric data, not only just descriptive approach. This approach represents a good direction because it allows readers to precisely define observed phenomenon, which is not an easy task. The characteristics of Silene seeds are particularly difficult to describe, due to their unique kidney shape. In the presented work, the authors performed a lot of detailed analyzes of the seed shape not only in various Silene species but also within the population, focusing on the dorsal view of seeds in the studied species.
In my opinion in the section “Materials and Methods” there is no information about the climatic conditions in which plants grew and formed seeds taken for the tests. Such information will allow to take into account environmental variability, which is very important in the case of plants.
Information has been added in relation to the growing conditions in the first paragraph of the Materials and Methods section:
“With the exception of S. latifolia 06, the collected seeds come from natural populations, each with its own particular environmental conditions.”
Besides, maybe it would be worth mentioning in the summary whether such models can also be used in the case of other plant species whose seeds have a kidney shape.
At the end of the summary it has been mentioned that the method is of more general application and can be applied to other species. The sentence:
“Based on the comparison with geometric models, a quantitative description of seed shape allows the study of variation in shape between species and in populations, as well as the identification of seeds in Silene species.”
Has been modified to:
”The quantitative description of seed shape based on the comparison with geometric models allows the study of variation in shape between species and in populations, as well as the identification of seeds in Silene species. The method is of application to other plant species.”
Once again, I think that it is interesting approach, worthwhile to be presented, but a few above minor points should be specified.
Thank you again for your comments.
Reviewer 2 Report
I read the manuscript ‘New geometric models for shape quantification of the dorsal view in seeds of Silene species’ with true interest. In general, the problem is stated plainly, the rationale of the analyses is more or less easy to understand. My first and overall impression is quite positive. But my reviewer’s task is to find weak points in the manuscript which should need to be corrected or discussed before the further editorial process. And in the case of this paper, interested and valuable, I would like to emphasise this, I found some issues that in my opinion should be clarified and corrected if possible. There are some questions however that worry me.
First, selection of the species. About 887 species belong to the genus Silene (according to POWO), while here only 12 were studied. Of course, I understand that not all could have been analysed, but 12 out of 887 give us only 1.4%. It is quite problematic to extrapolate the results to a larger number of taxa if only such a small fraction of species was studied. Second, what was the criterion for selecting this set of species? I think it should be explained in the text.
Let’s skip to the statistical questions and worries. I have two major concerns.
Normality of the data distribution is always a problem. Here, we have 26 populations and 7 traits (for area, perimeter, etc., see tables 1 & 2), which gives us 182 statistical populations. The authors provide the information saying that ‘it cannot be rejected that the data came from a normally distributed population’. This is unclear for me. It should be ‘populations’, not ‘population’. For normality, all 182 should be tested (and meet this condition), otherwise, ANOVA cannot be performed. I would like to emphasise: all the statistical populations should be tested against normality and variance homogeneity before ANOVA. This concerns me a lot. Especially, when I realised that Silene gallica has mixed types of seeds and because table A1 presents a quite wide range of values of variation coefficients. From my experience, it is quite impossible to satisfy this condition for such a large set of data. I would like the authors to present the tests’ results in Supplementary Materials. The same should be applied if species are compared, also for other traits, such as the J index.
One of the major questions I asked myself when reading the paper was: why two groups of taxa were analysed separately (for convex and non-convex seeds)? What was the reason for that decision? In my view, all the populations/species should be analysed together. This would make the study much easier to understand and read. Especially in light of S. gallica that was analysed in both groups. Why not pull all the species into common analyses?
This type of data seems to me ideal for Elliptic Fourier Analysis. What about including this topic at least in the Discussion?
Other minor remarks:
In the M&M section, I did not find any information about Tuckey’s test which supposedly was performed (according to tables 2, 3, 4 & 5). Only Scheffé test is mentioned there.
lines 68-85: In my view, this fragment doesn’t fit well here. I suggest moving it to the Discussion section. Here, it makes the reading difficult as it disrupts the flow of the Introduction.
Figure 1. The outline of DM2 definitely seems to me to be non-convex.
I am not a native speaker, so maybe my opinion is not well-founded, but I must say I liked the English of the manuscript. I detected only minor issues, like ‘straights’ (l. 419) or ‘an-other’ (l. 528).
Author Response
Dear Reviewer 2:
Thank you for your commentaries that have contributed to describe more precisely the characteristics of the seed populations and to improve notably the quality of the article.
Changes have been made according to your instructions as indicated below (in black are your commentaries; in red, our responses and new annotations in the article):
I read the manuscript ‘New geometric models for shape quantification of the dorsal view in seeds of Silene species’ with true interest. In general, the problem is stated plainly, the rationale of the analyses is more or less easy to understand. My first and overall impression is quite positive. But my reviewer’s task is to find weak points in the manuscript which should need to be corrected or discussed before the further editorial process. And in the case of this paper, interested and valuable, I would like to emphasise this, I found some issues that in my opinion should be clarified and corrected if possible. There are some questions however that worry me.
First, selection of the species. About 887 species belong to the genus Silene (according to POWO), while here only 12 were studied. Of course, I understand that not all could have been analysed, but 12 out of 887 give us only 1.4%. It is quite problematic to extrapolate the results to a larger number of taxa if only such a small fraction of species was studied. Second, what was the criterion for selecting this set of species? I think it should be explained in the text.
This article represents a first approach to the morphological analysis of the dorsal views of Silene seeds based on geometric models. Thus, we did not pretend to extrapolate the results to a majority of species of Silene. The objective is to show that the method of shape quantification by comparison with geometric models is useful to describe, quantify and compare seed shape between different species, both in the lateral, as well as in the dorsal views. The species were chosen to confirm the hypothesis that there are differences in shape between both subgenera, Silene and Behenantha, as it was reported before.
In consequence, the main criterion was to have included in the study species from the two major subgenera, Silene subg. Silene and S. subg. Behenantha and with different shape in their dorsal views.
To explain the species analysed, the following statement is now included at the end of the first paragraph of Materials and Methods:
“The species were chosen to investigate differences in shape between both subgenera, Silene and Behenantha, in support of results reported before and to explore in detail these differences [18,20].”
Let’s skip to the statistical questions and worries. I have two major concerns.
Normality of the data distribution is always a problem. Here, we have 26 populations and 7 traits (for area, perimeter, etc., see tables 1 & 2), which gives us 182 statistical populations. The authors provide the information saying that ‘it cannot be rejected that the data came from a normally distributed population’. This is unclear for me. It should be ‘populations’, not ‘population’. For normality, all 182 should be tested (and meet this condition), otherwise, ANOVA cannot be performed. I would like to emphasise: all the statistical populations should be tested against normality and variance homogeneity before ANOVA. This concerns me a lot. Especially, when I realised that Silene gallica has mixed types of seeds and because table A1 presents a quite wide range of values of variation coefficients. From my experience, it is quite impossible to satisfy this condition for such a large set of data. I would like the authors to present the tests’ results in Supplementary Materials. The same should be applied if species are compared, also for other traits, such as the J index.
Thank you for pointing to these important questions.
The original data set used in statistical calculation has been made available at: https://zenodo.org/record/6276242#.YhiliJaCGUk.
The box-plot representations corresponding to the populations under study are available at: https://zenodo.org/record/6298757#.Yhn4TujMKM8
And this is now indicated at the beginning of the section 4.5. Statistical analysis.
Including the measurements of J index, we have a total of 291 statistical populations. In response to your commentary, the following sentence in the Materials and Methods section 4.5 (Statistical analysis) has been deleted:
According to Kolmogorov and Shapiro-Wilk tests, it cannot be rejected that the data came from a normally distributed population.
And a detailed explanation is now given as follows:
“Data from table 1, 2 and 3 did not follow a normal distribution, and were transformed according to Wessa [40]. The results of Shapiro-Wilk normality test for each statistical analysis can be found at: https://zenodo.org/record/6325113#.YiCR3ujMKM8.
The broad range in variation coefficients concerns all the measurements, the range is much more restricted when looking separately at each measurement.
One of the major questions I asked myself when reading the paper was: why two groups of taxa were analysed separately (for convex and non-convex seeds)? What was the reason for that decision? In my view, all the populations/species should be analysed together. This would make the study much easier to understand and read. Especially in light of S. gallica that was analysed in both groups. Why not pull all the species into common analyses?
Two groups of taxa for convex and non-convex seeds were analysed separately because the models are quite different. While convex models form a continuity, and there is not a straight relationship one model-one species, non-convex models are more species specific. Values of J index obtained with one model in species other than the one adjusting to this model are very low. This has been now indicated in the following way:
“Two groups of taxa for convex and non-convex seeds were analysed separately be-cause the models are quite different. While convex models form a continuity, and there is not a straight relationship one model-one species, non-convex models are more species specific. Values of J index obtained with one model in species other than the one adjusting to this model are very low.”
This type of data seems to me ideal for Elliptic Fourier Analysis. What about including this topic at least in the Discussion?
This topic has now been included in the discussion.
Other minor remarks:
In the M&M section, I did not find any information about Tuckey’s test which supposedly was performed (according to tables 2, 3, 4 & 5). Only Scheffé test is mentioned there.
Tuckey’s test is now mentioned in this section.
lines 68-85: In my view, this fragment doesn’t fit well here. I suggest moving it to the Discussion section. Here, it makes the reading difficult as it disrupts the flow of the Introduction.
This paragraph has been moved to the discussion
Figure 1. The outline of DM2 definitely seems to me to be non-convex.
Please see the definition of convex in rows 78-81:
“In this work, convexity is referred to a particular feature of the planar image of the dorsal view of the seed. The corresponding region is said to be convex when the straight line joining any pair of internal points lies entirely inside the region.”
I am not a native speaker, so maybe my opinion is not well-founded, but I must say I liked the English of the manuscript. I detected only minor issues, like ‘straights’ (l. 419) or ‘an-other’ (l. 528).
These issues have been corrected
Reviewer 3 Report
The manuscript of Rodriguez-Lorenzo and co-authors entilted New geometric models for shape quantification of the dorsal view in seeds of Silene species, present morphological classification of seed silhouettes. The manuscript is based on a large sample of seed assetions with several population and speciees. However, it is pretty tedious to follow, partly because as it is mainly methodological and the method section is at the end, but also because some sentences are very long and need some English editing. The paper describe some shape matching techniques based on diverse geometrical models. I agree that this approach update the purely descriptive approach of the XIX century taxonomists. However, it creates in my opinion deep artificial discreteness between morphological types. Why not using Fourier analysis which is very suited to describe such outlines and to quantify distances between two shapes? Using such discrete descriptors, make that the final assertion of the discussion (finding QTL underlying shape differences) will likely be unsuccessful. Beside this personal feeling that shape is a multivariate continuous trait and we have to be very cautious in trying to individualize morphotypes and analyze them, I have also some concerns about the employed methodology.
First, the alignment of seed outline is likely very important in the outcomes but it is not explained. It appears very manual to me without strict control. Similarly for the technique of extracting an average silhouette. Using Fourier transform for example will have provide a more consistent way of defining the average. From which you still can try to describe in term of morphotypes (= geometric models) but keeping information about variability and between type variation and be able to analyze them. For instance, the authors spend a lot of text to describe the matching to each type when a multivariate analysis will have provided a more readable and complete information. It will really provide why the authors promise at the end of the introduction "give for the first time, data on variation of seed shape in populations"
Also, why using this J-index? In my opinion, root mean squared error (RMSE) is a more standard index use in shape matching like in ICP matching.
In conclusion, the manuscript has some merits but needs important clarification and simplification.
Author Response
Dear Reviewer 3:
Thank you for your commentaries that have contributed to improve the quality of the article.
Changes have been made according to your instructions as indicated below (in black are your commentaries; in red, our responses and new annotations in the article):
The manuscript of Rodriguez-Lorenzo and co-authors entilted New geometric models for shape quantification of the dorsal view in seeds of Silene species, present morphological classification of seed silhouettes. The manuscript is based on a large sample of seed assetions with several population and speciees. However, it is pretty tedious to follow, partly because as it is mainly methodological and the method section is at the end, but also because some sentences are very long and need some English editing. The paper describe some shape matching techniques based on diverse geometrical models. I agree that this approach update the purely descriptive approach of the XIX century taxonomists. However, it creates in my opinion deep artificial discreteness between morphological types.
The commentaries above raise interesting questions. First, it is indicated that the method presented creates deep artificial discreteness between morphological types. Our experience is that, at least in some instances, there is not a total continuity, and in consequence it is licit to talk of discrete morphological types. In fact, the morphological types shown in the non-convex seeds are quite species specific.
We agree that in many instances, shape, similar to size, may be a multivariate continuous trait. Nevertheless, in other instances, some aspects of seed shape may be due to particular genetic determinants.
The results presented for the non-convex seeds indicate different morphotypes in different species, making the approach to finding QTL underlying shape differences quite reasonable.
Why not using Fourier analysis which is very suited to describe such outlines and to quantify distances between two shapes? Using such discrete descriptors, make that the final assertion of the discussion (finding QTL underlying shape differences) will likely be unsuccessful. Beside this personal feeling that shape is a multivariate continuous trait and we have to be very cautious in trying to individualize morphotypes and analyze them, I have also some concerns about the employed methodology.
In relation with Fourier analysis, we agree that the data here presented adjust quite well to this type of study, and that it also results in quantification of shape. Nevertheless, in contrast to the method here presented, Fourier analysis does not allow directly to identify every shape with a well defined geometric figure. In comparison with elliptic Fourier analysis, the method here presented compares bi-dimensional seed shape with a figure of reference, thus the results are not purely numerical or statistical, but analytical, providing geometric models with their corresponding equations. A high value in J index means that the seeds of a species have a given shape, defined by the similarity to a model; and the difference detected between two species or varieties as the result of a statistical test means that there are morphological differences related with a different degree of similarity to a given model between the species or varieties tested. Thus, the results of a statistical test are compared directly with visual information given by the models.
The following paragraph has been added to the discussion:
Fourier analysis has been applied to seed shape quantification in a number of plant species [27-30]. The experimental setup presented in this work compares bi-dimensional seed shape with a figure of reference, thus the results are not only numerical and statistical but analytical, which provides geometric models with their corresponding equations. In consequence, the method here reported presents a complementary approach to Fourier analysis providing the similarity of a seed image with a given geometric figure as a measurement that can be submitted to statistical comparison.
First, the alignment of seed outline is likely very important in the outcomes but it is not explained. It appears very manual to me without strict control. Similarly for the technique of extracting an average silhouette. Using Fourier transform for example will have provide a more consistent way of defining the average. From which you still can try to describe in term of morphotypes (= geometric models) but keeping information about variability and between type variation and be able to analyze them. For instance, the authors spend a lot of text to describe the matching to each type when a multivariate analysis will have provided a more readable and complete information. It will really provide why the authors promise at the end of the introduction "give for the first time, data on variation of seed shape in populations"
A multivariate analysis has been now provided in the results section.
Also, why using this J-index? In my opinion, root mean squared error (RMSE) is a more standard index use in shape matching like in ICP matching.
J-index gives a direct measure of the similarity of the seed with a geometric figure well defined. It permits to associate the shape of seeds with known geometric figures and to compare between species and varieties.
In conclusion, the manuscript has some merits but needs important clarification and simplification.
We thank you for the review and expect to have made steps forward in the desired simplification.
have contributed to reach the desired simplification.
Round 2
Reviewer 2 Report
I did my best to re-read the manuscript with the changes applied by the authors. I must say it has been significantly improved. Besides, I really appreciate that the data are available now.
Please, do not take me wrong, but I still have some doubts, even if the response was quite satisfactory. What concerns me here (still) is the faultlessness of the analyses. In my view, the study is interesting, I do not doubt it. However, it should be supported clearly and impeccably by adequate statistics. Otherwise, it may lose its power and soundness. My point is that the procedure of how the analyses of ANOVA were performed is not clear for me. I would like the authors to clear up my doubts and hesitations.
In the data I downloaded (let’s focus here only on the data from tables 1 and 2), I detected that 30 sets out of 182 do not meet the assumption for normality of distribution. Many populations do not have equal variances either (please find it in the pdfs I am attaching to this review). It means that ANOVA could not be performed straightaway. Some additional procedures, like transformations, should be done first. Or one can resort to non-parametric methods.
If I had had the original data, I would have checked if all 182 statistical populations met the assumptions for AVOVA, i.e. normal distribution and if variances are equal. If not, ALL the data for each taxon for a particular trait in which the assumptions were not met, should be transformed and CHECKED AGAIN. I would like to emphasise it: all the data for a particular trait, not only which did not meet the assumptions. This is crucial. In table 1 for the trait ‘area’ 2 out of 17 populations do not meet the assumption about normal distribution. It means that all 17 datasets for this trait should be transformed and checked again. If after such a procedure, it turns out that the data for the trait can not be used for ANOVA, I would opt for a non-parametric test.
I simply fear that the ANOVAs and respective Tukey’s tests were performed without meeting the initial assumptions. There are a lot of outliers and very non-normal distributions among the datasets, not to mention that in many cases variances are not equal.
As far as I can understand from the correction to the text, the authors used the Box-Cox transformation. It is difficult to check its performance without additional parameters (lambda). From the data I downloaded, i.e. test results, it is not clear however if the results are for the datasets before or after the transformation. If they were done after transformation, it means that many populations still do not have normal distribution thus ANOVA was not legitimate. If however, the presented results of the tests were done before the transformation, it would be highly recommendable to present the results of the same tests after the transformation. The same concerns the tests for variances.
In any case, there is still possible to perform non-parametric analyses.
I would like to be convinced my doubts are baseless. I would be more than happy to see the paper published!
Here I am putting 'major revision'. But if am wrong, my suggestion is 'minor', and only more precise description of the methods would be necessary.

Author Response
Dear Reviewer 2,
Thank you very much for your commentaries to the article. Following your recommendation and after carefully reviewing the data, we decided to do non-parametric tests. In addition, the data have been re-ordered and the results are presented by species. In consequence, Table 4 has been deleted.
Tables 1 to 3 have been elaborated with the results of Kruskal Wallis tests applied to the samples, followed by stepwise stepdown comparisons as post-hoc tests (Campbell and Skillings, 1985). The corresponding changes have been made to the Materials and Methods section and the reference indicated has been added to the reference list.
The data corresponding to Silene gallica have been left in Tables 1 and 3, corresponding to the higher results obtained in this species with the convex model (DM2). Thus, according to your commentary, Silene gallica is now presented in the group of convex seeds.
We want to thank you for your contribution to improve the presentation of the results.
Round 3
Reviewer 2 Report
I am really glad the Authors made the changes in the manuscript opting for non-parametric methods, which in this case, seem to be a safer way and much more accurate.
The only thing I would like to suggest here is that the Authors mention what exactly post hoc tests they used.
Secondly, please note that the term 'Post Hoc' was used together with 'post-hoc'. I think the form 'post hoc' (small letters, no dash) is preferred.
Author Response
Dear Reviewer 2,
Following your recommendation, the post hoc tests used have now been indicated in the corresponding section of Materials and Methods (4.5. Statistical analysis).
The form 'post hoc' (small letters, no dash) is now used in all instances.
We thank you very much for your contribution to the fianl version of the article.